# OpenReview forum: "FoPO: Foresight Policy Optimization Incentivizes Strategic Reasoning in LLMs"
_ICLR.cc/2026/Conference — ICLR 2026 Conference Withdrawn Submission_

### Official Review · Reviewer_HARK · 2025-10-28

**Soundness:** 2
**Presentation:** 2
**Contribution:** 2
**Rating:** 2
**Confidence:** 4

**Summary:**

This work aims to improve the foresight ability of LLM agents in strategic environments. To do this, the authors first curated two new datasets in one cooperative game and one competitive game, then they proposed the FoPO algorithm that extend PPO by considering the learning dynamics of other agents. The LLM is trained by first imitation learning and then offline RL on the curated dataset. The experiment results show the proposed method perform better than imitation learning, PPO, and ArCHer in training games and generalize to unseen games.

**Strengths:**

1. Clear motivation: forecasting other agents' behaviors is a distinct and important problem in multi-agent learning.
2. Straightforward method: the idea of the proposed method is straightforward and well-explained. More concretely, FoPO extends PPO by considering the learning dynamics of the other agent and adds a new term in the gradient.

**Weaknesses:**

1. Online algorithm with offline data: the proposed algorithm is built on PPO, an online RL algorithm, yet it is trained using an offline dataset. PPO’s clipped objective and advantage estimation inherently assume that samples are drawn from the current (or near-current) policy. Applying PPO directly to an offline dataset thus violates this assumption and leads to a severe distributional mismatch between the fixed behavior policy and the evolving target policy. Prior studies in offline RL [1, 2] have demonstrated that such online RL algorithms fail to provide reliable learning signals in purely offline settings unless off-policy bias is corrected or the learned policy is explicitly constrained.
2. Almost the same algorithm has been proposed in prior work: LOLA [3], a well-known work on opponent shaping published in 2017, has proposed almost the same algorithm as FoPO. More concretely, Eq. (4.2) in LOLA is
    $$V^{1}(\theta^{1}, \theta^{2} + \Delta \theta^{2}) \approx V^{1}(\theta^{1}, \theta^{2}) + (\Delta \theta^{2})^{\top} \nabla_{\theta^{2}} V^{1}(\theta^{1}, \theta^{2}),$$
    and Eq. (7) in FoPO is
    $$O_1(\theta, \theta + \Delta\theta) \approx O_1(\theta, \theta) + (\Delta\theta)^{\top} \nabla_{\theta_2} O_1(\theta, \theta_2) \Big|_{\theta_2 = \theta},$$
    which are exactly the same. The only difference is that LOLA is built on naive policy gradient while FoPO is built on PPO, which diminishes the novelty of the proposed algorithm.
3. Limited improvement in out-of-domain generalization: in Table 2, the average improvement of FoPO is less than 1% in 5 out of 6 settings compared to ArCHer. Moreover, if we compare the performance of IM in Table 1 and FoPO in Table 2, the average improvement is also around 2%, which is marginal.

[1] Levine, Sergey, et al. "Offline reinforcement learning: Tutorial, review, and perspectives on open problems." arXiv preprint arXiv:2005.01643 (2020).

[2] Kumar, Aviral, et al. "Conservative q-learning for offline reinforcement learning." Advances in neural information processing systems 33 (2020): 1179-1191.

[3] Foerster, Jakob, et al. "Learning with Opponent-Learning Awareness." International Conference on Autonomous Agents and Multiagent Systems (AAMAS), 2017.

**Questions:**

1. (W1) How to solve the problem of distributional mismatch when using online RL algorithms on offline datasets?
2. (W2) What is the novelty of FoPO compared to existing opponent shaping work like LOLA?
3. What is the opponent model for evaluation in Figure 4? Do you have results between every two models like in Figure 3?
4. Does FoPO train a critic? If so, it would be better to provide more training details, like whether the actor and critic share the same backbone, the value loss and coefficients. If not, how do you calculate the advantage in PPO and FoPO?

---

### Official Review · Reviewer_12Bu · 2025-10-29

**Soundness:** 2
**Presentation:** 2
**Contribution:** 2
**Rating:** 2
**Confidence:** 4

**Summary:**

The paper proposes Foresight Policy Optimization (FoPO), a new reinforcement learning algorithm that enhances strategic reasoning in LLMs by allowing agents to anticipate and influence their counterparts’ future actions.

FoPO extends PPO with a foresight correction term using coupled gradients between agents.
The authors also curate two game-theoretic datasets — Cooperative RSA (for collaboration) and Competitive Taboo (for competition) — and show through experiments on Meta-Llama-3-8B and Qwen3-14B that FoPO improves both cooperative and competitive reasoning, outperforming PPO and other baselines on γ-Bench and in-domain tasks.

**Strengths:**

The paper presents a creative and well-structured experimental study. Its main strength lies in the novel idea of incorporating foresight into PPO to model mutual influence between agents, and in the carefully designed experiments, which comprehensively validate the method through both cooperative and competitive settings.

**Weaknesses:**

* First, the paper clearly lacks sufficient polishing in writing and formatting. For example, Equations (2) and (3) should be presented as a single equation rather than two separately numbered ones. In addition, the outer parentheses in Equation (11) are improperly formatted and should use \left( \right) for consistency and readability.

---

* In terms of research content, while the proposed idea (insight) shows some originality, the overall reasoning and methodological design are not sufficiently rigorous. The authors overlook a key issue — **why the proposed FoPO modification is suitable for large language models (LLMs)**. After reading the entire paper, I found no substantive analysis explaining how the algorithm aligns with the specific characteristics or training dynamics of LLMs. In contrast, prior works such as GRPO [1] systematically examine the structural and optimization properties of LLMs to justify their design choices. The improvement proposed in this paper, however, is almost entirely grounded in traditional game-theoretic or multi-agent reinforcement learning (MARL) perspectives. If positioned this way, the authors should instead perform systematic comparisons in well-established multi-agent environments such as poker or StarCraft, where standardized baselines (e.g., MAPPO, QMIX) and mature training pipelines exist. This would allow for a clearer and fairer evaluation of the method’s advantages and limitations in conventional MARL settings.

---

* Moreover, the algorithmic design only considers one-step foresight in two-player games, which is straightforward and lacks theoretical depth. From the standpoint of game theory concepts such as common knowledge and bounded rationality, this one-step foresight can only be viewed as a simplified special case. Extending the strategy gradient update to incorporate multi-step foresight over $N$ interactions among $K$ agents—analogous to the GAE formulation—would make the framework more complete and potentially yield substantial performance improvements. If computational constraints specific to LLM training motivated the one-step simplification, this rationale should be clearly articulated in the paper.

---

* Finally, the absence of comparison with GRPO is an omission. GRPO has consistently shown superior performance to PPO across various settings and is widely recognized as a key baseline for LLM post-training. Failing to include it in the experiments substantially undermines the credibility and impact of the claimed contributions.

[1] Shao Z, Wang P, Zhu Q, et al. Deepseekmath: Pushing the limits of mathematical reasoning in open language models[J]. arXiv preprint arXiv:2402.03300, 2024.

**Questions:**

Refer to the previous section

---

### Official Review · Reviewer_1yAq · 2025-10-31

**Soundness:** 2
**Presentation:** 2
**Contribution:** 3
**Rating:** 4
**Confidence:** 4

**Summary:**

The authors propose Foresight Policy Optimization (FoPO), an extension of PPO that introduces a foresight term to explicitly model how the agent’s current action at time t influences the opponent’s response at time t+1, and consequently affects future rewards. In addition to the algorithmic contribution, the authors construct two game-based dialogue benchmarks—Cooperative RSA and Competitive Taboo—to evaluate strategic reasoning in both cooperative and adversarial settings. They also curate a high-quality dataset to capture these two reasoning dimensions. Experimental results on the γ-bench show that FoPO consistently outperforms baseline methods such as ICL, PPO, and ArCHer across multiple model backbones (Llama3-8B, Qwen3-14B), achieving both higher performance and faster convergence.

**Strengths:**

- The paper introduces a differentiable and trainable coupling mechanism that explicitly models the opponent’s foresight update within the PPO framework. The idea is clear, novel, and provides a strong baseline for subsequent studies on strategic reasoning through reinforcement learning.
- The authors construct two well-curated datasets—Cooperative RSA and Competitive Taboo—each designed to represent cooperative and competitive dimensions of strategic reasoning, offering a reliable testbed for future research.

**Weaknesses:**

- Although many studies have pointed out that *Offline PPO* often fails to perform effectively, this paper reports performance gains using *Offline PPO/FoPO* without providing further justification or analysis of why the method succeeds in this context.
- PPO typically involves two key networks—the actor and the critic—but the paper omits any discussion of the critic. More details are needed on its design, its relationship to the actor, and the coefficients used for actor and critic losses.
- Certain expressions and notations are unclear or inconsistent: for instance, *imitation learning* should be abbreviated as **IL** rather than **IM**; and in Figure 3, the reward range exceeds 50, which conflicts with the normalization assumption (expected range [0,1]).

**Questions:**

1. In the γ-bench evaluation, which specific scenarios contribute most to the overall performance gain? Could the authors provide representative failure cases and error type analysis? Has the generalization of FoPO been validated on broader or more challenging tasks?
2. In Equation (11), when ( T = n ), the numerator becomes 0, leading to ( R = 1 ); when ($ T = \text{Conv}_{\min} $), the term equals 1, yielding ( R = 0 ). This seems counterintuitive—could the authors clarify this formulation?
3. Why does *ICL* require parameter updates? Typically, in-context learning (ICL) belongs to the *prompt engineering*paradigm and does not involve gradient updates.
4. The comparison between IL and FoPO shows only a 1% improvement in OOD settings. Since RL methods are generally expected to outperform IL in out-of-distribution generalization, does this imply that the current RL design may need further refinement?

---

### Official Review · Reviewer_vTu3 · 2025-11-01

**Soundness:** 2
**Presentation:** 2
**Contribution:** 2
**Rating:** 2
**Confidence:** 4

**Summary:**

The paper proposes Foresight Policy Optimization multi-agent LLMs to anticipate how an opponent will update after the agent’s move. Concretely, FoPO adds a coupled-gradient term derived from a first-order Taylor expansion of the opponent’s next update, yielding an extra factor that depends on how the agent’s current policy affects the opponent’s future advantage (and vice versa). To showcase benefits, the authors curate two synthetic interaction datasets: Cooperative RSA (speaker/listener identify a target object in minimal turns; reward depends on distance from the minimal conversation length) and Competitive Taboo (attacker tries to induce a target word; defender avoids/identifies). They train Llama-3-8B-Instruct and Qwen3-14B with imitation, PPO, ArCHer, and FoPO, and evaluate both in-domain and on γ-Bench.

**Strengths:**

1. The work targets a less-explored question, i.e., strategic reasoning that differs significantly from single-agent reasoning.
2. Apart from the algorithmic contributions, the paper also introduced new benchmarks, Cooperative RSA and Competitive Taboo isolating distinct facets (speaker/listener pragmatics vs. zero-sum deception/defense), which is useful for controlled analysis
3. On several setups, FoPO outperforms PPO/ArCHer and mitigates some reward-hacking pathologies reported for PPO on Taboo

**Weaknesses:**

1. The core idea of the paper is anticipating an opponent’s learning step and compute gradient against the anticipated opponent instead of the current opponent. However, I believe this is a well-established idea in the area of multi-agent RL and game theory, i.e., opponent-learning-aware methods (e.g., LOLA-style meta-gradients). The paper neither situates FoPO rigorously against that line nor compares empirically, leaving originality unclear. It seems to me that the so-called FoPO is just an implementation of this idea in the LLM domain. Meanwhile, this line of literature is not mentioned.

2. Specifically, the derivation of the FoPO is not rigorous. The gradient expression of the desired objective should involve a second-order gradient term. However, this is not the case of the gradient in the main paper. Does the paper make some non-trivial simplifications to make the objective first-order? Or the paper just relies on second-order gradient? I believe either case requires in-depth theoretical analysis.

3. The choice of baselines also requires justifications. Since the paper introduces some game-theoretical learning objectives for LLM training, shouldn't the paper compares with some other game-theoretical/minimax optimization algorithm? Like classical extra-gradient, optimistic gradient, or modern ones tailored for deep learning, such as opponent shaping line of literature.


4. When studying opponent-aware/anticipation style algorithms, it makes more sense to consider ``general-sum'' game, not zero-sum or fully cooperative one.

5. It seems to be the performance increase compared with PPO and ICL is not that consistent and sometimes marginal

**Questions:**

See above.

---

### Author Response · Authors · 2025-11-15
**Overall Response**

We sincerely thank the AC and reviewers for their thoughtful feedback. We found the comments highly valuable for improving the clarity and positioning of our work. This rebuttal focuses on two key points: (i) clarifying the contributions and positioning of FoPO, and (ii) addressing specific questions raised by the reviewers. We hope the following responses resolve the concerns, and we appreciate any further suggestions.

### **Position and Contribution**

We position our work as an NLP application aimed at enhancing strategic reasoning in LLMs, and we selected the corresponding track (not RL) accordingly. The primary contribution therefore lies in our task insight, instead of solution/dataset/training insight, and we carefully chose methods and datasets specifically to address the subtasks involved. In our work, the training method (self-play multi-agent training), the dataset, and the modeling approach all serve the goal of evaluating whether an LLM with enhanced strategic reasoning capabilities can be trained effectively.

The paper's position determines the comparative methods and dataset. We believe that the choice of comparative methods should be guided by two considerations: (1) suitability for interaction, and (2) relevance to enhancing LLM reasoning. Accordingly, we think our work should be compared with previous fine-tuning methods for reasoning (e.g., SFT/imitation learning, PPO and ArCHer), rather than the multi-agent methods suggested by reviewers. We mentioned multi-agent settings because strategic reasoning is inherently linked to such environments; within these settings, we selected Cooperative RSA and Competitive Taboo among many possible tasks. However, it is not true that the studies of strategic reasoning are those of multi-agent learning. If reviewers did not question our dataset selection, it is unclear why the method choice should be judged differently.

For training an LLM with enhanced strategic reasoning, our key contributions are summarized as follows:

Foresight Policy Optimization (FoPO): We propose FoPO, a novel extension of PPO that incorporates coupled gradients during parameter updates. FoPO enables LLM agents to anticipate and strategically influence the future actions of counterpart agents within multi-agent environments, thereby maximizing expected rewards for strategic reasoning.

High-quality dataset: We curate a dataset comprising game-theoretic tasks, Cooperative RSA and Competitive Taboo, which capture two distinct aspects of strategic reasoning. This dataset enables LLMs to learn strategies more effectively.

Comprehensive evaluation: We conduct extensive experiments using the γ-bench benchmark to evaluate FoPO. Our results show that FoPO not only significantly enhances LLMs’ strategic reasoning abilities but also accelerates policy convergence.


### **Questions About FoPO**
We summarize the questions and would like to answer them one by one.

1. **Novelty and positioning relative to existing opponent-aware methods.** FoPO is conceptually related to opponent-aware approaches such as LOLA and CoPO. We initially attempted to directly apply these methods in our LLM-based self-play environment and have cited them in the submitted materials; full citations will be added in the final version. While some of our derivations resemble these prior formulations, this similarity is expected: anticipating the opponent's update direction has become a standard modeling principle in the literature.

One of our core distinctions lies in computational design. FoPO is intentionally constructed to require **only first-order computations**, which is crucial for training large language models where second-order derivatives impose prohibitive memory and computational costs. This makes direct application of second-order methods like LOLA or CoPO practically infeasible in LLM-scale optimization. In addition, prior work focuses primarily on general-sum interactions as stated by Reviewer vTu3; FoPO extends opponent-aware optimization to **cooperative settings**, enabling strategic-reasoning learning in scenarios that have been less explored in earlier formulations.

2. **Notations and equations.** We appreciate the reviewers’ detailed feedback regarding notation and expression. We will revise the main text and figures to ensure terminology is used consistently, including clarifying the definition of Imitation Learning and correcting the normalization range of rewards as suggested.

3. **Motivation of the method.** Our design is **problem-driven rather than method-driven**. FoPO is tailored to the constraints of LLM optimization, including the need for computational efficiency and training stability. However, existing opponent-aware algorithms do not directly address. These considerations, and their implications for designing a scalable optimization procedure, are already discussed in the introduction and method sections and will be clarified further in the revision.

---

### Note · Authors · 2026-01-03

I have read and agree with the venue's withdrawal policy on behalf of myself and my co-authors.